# The Effect of Low Doses of Acetylsalicylic Acid on the Occurrence of Rectal Aberrant Crypt Foci

**DOI:** 10.3390/medicina58121767

**Published:** 2022-11-30

**Authors:** Marek Kowalczyk, Dawid Sigorski, Łukasz Dyśko, Ewa Zieliński, Dana Zupanovich Lucka, Łukasz Klepacki

**Affiliations:** 1Department of Psychology and Sociology of Health and Public Health, University of Warmia and Mazury, 10-719 Olsztyn, Poland; 2Clinic of Oncological and General Surgery, University Clinical Hospital in Olsztyn, 11-041 Olsztyn, Poland; 3Department of Oncology, Collegium Medicum, University of Warmia and Mazury, 10-719 Olsztyn, Poland; 4Department of Emergency Medicine, Collegium Medicum in Bydgoszcz, Nicolaus Copernicus University, 85-067 Bydgoszcz, Poland; 5Department of Business, Fresno Pacific University, Fresno, CA 93702, USA; 6Department of Anatomy, University of Warmia and Mazury, 11-041 Olsztyn, Poland

**Keywords:** rectal aberrant crypt foci (ACF), colorectal adenoma, colorectal cancer (CRC), acetylsalicylic acid (ASA), colonoscopy

## Abstract

*Background and Objectives*: Aberrant crypt foci (ACF) are one of the earliest putative preneoplastic and, in some cases, neoplastic lesions in human colons. Many studies have confirmed the reduction of ACFs and colorectal adenomas after treatment with acetylsalicylic acid (ASA) commonly referred to as ASA; however, the minimum effective dose of ASA and the duration of use has not been fully elucidated. The objective of our study was to assess the significance of low dose ASA (75-mg internally once daily) to study the chemopreventive effect of ASA in ACF and adenomas development in patients taking this drug for a minimum period of 10 years. *Materials and Methods*: Colonoscopy, combined with rectal mucosa staining with 0.25% methylene blue, was performed on 131 patients. The number of rectal ACF in the colon was divided into three groups: ACF < 5; ACF 5–10; and ACF > 10. Patients were divided into two groups: the “With ASA” group (the study group subjects taking ASA 75-mg daily for 10 years); and “Without ASA” group (control group subjects not taking ASA chronically). The incidence of different types of rectal ACF and colorectal polyps in both groups of subjects was analysed and ascertained. *Results*: Normal ACF was found in 12.3% in the study group vs. 87.7% control group, hyperplastic 22.4% vs. 77.6%, dysplastic 25% vs. 75%, mixed 0% vs. 100%. Treatment with ASA affects the occurrence of colorectal adenomas. The amount of dysplastic ACFs was lower in the study group than in the control group. The increase in dysplastic ACFs decreases with age in both groups, with the increase greater in those not taking ASA. *Conclusions*: Patients who take persistent, chronic (>10 years) low doses of ASA have a lower total number of all types of rectal ACFs and adenomas compared to the control group.

## 1. Introduction

Despite the rapid introduction of various diagnostic and treatment modalities, colorectal cancer (CRC) remains one of the most common causes of cancer deaths worldwide. According to The Surveillance Epidemiology and End Results Program [1], in 2022, it is estimated that 52,580 people will die of colorectal cancer. Aberrant crypt foci (ACF) are one of the earliest putative preneoplastic and, in some cases, neoplastic lesions in human colons. There have been many studies on the use of different agents in the chemoprevention of CRC precursors (ACF, adenomas): calcium supplements [2], Cyclooxygenase-2 (COX-2) inhibitors [3], or other non-steroidal anti-inflammatory drugs (NSAIDs) [4,5]. Several studies have also confirmed the close association of rectal ACF with adenomas and CRC [6,7,8]. Experimental studies have shown a reduction in the number of ACFs after treatment with acetylsalicylic acid (ASA), NSAIDs and COX-2 inhibitors [5,9,10,11]. Many scientific reports have addressed the critical role of COX-2 overexpression and concomitant overproduction of prostaglandin, which may play an essential role in impaired apoptosis, cell proliferation, and abnormal vascularization of developing CRCs precursors [12,13,14,15,16]. ASA preparations, in contrast, play a significant role in preventing adverse cardiovascular events, such as myocardial infarction, coronary artery disease exacerbations, and cerebrovascular incidents [17]. Therefore, a significant population of patients take ASA as a chemotherapeutic agent for cardiovascular events. Despite over 30 years of research, many aspects of this drug are still unclear, and the lowest chemopreventing dose of ASA in the formation of ACF and adenomas has not been established. Moreover, it has not been fully evaluated as to whether the local or systemic action of ASA on the colonic mucosa is significant [18]. Another important aspect is the bioavailability of the different drug forms [19]. Enteric tablets are more bioavailable to the mucosa of the large intestine than fast dissolving preparations. ACF as small mucosal changes (up to 200 intestinal crypts in the focus) can be very susceptible to ASA available in the mucosa, as well as to its form absorbed in the gastrointestinal tract and present in the blood serum.

The aim of our research was to study whether the long-term (over 10 years) intake of low doses of acetylsalicylic acid (75-mg) in the enteric form has an impact on the occurrence of rectal ACF and colorectal adenomas, because this form and dose of ASA is routinely prescribed for patients with cardiovascular diseases.

## 2. Materials and Methods

The study was approved by the Bioethics Committee at the Faculty of Medical Sciences of the University of Warmia and Mazury in Olsztyn—Resolution No. 11/2010 29 April 2010. Each of the study participants gave informed consent to participate in the study. An examination was performed on patients referred by a general practitioner, according to routine indications. Colonoscopy, combined with rectal mucosa staining with 0.25% methylene blue was performed in 131 patients. Methylene blue stains the intestinal crypts blue, thus making them easier to identify. Three bioptates were collected from the foci defined macroscopically as ACF. The number of collected foci was respectively lower in cases where there were fewer foci. Additionally, before the colonoscopy, patients completed a questionnaire regarding epidemiological data that were used for the analysis of factors affecting the occurrence of ACF in the study group. Patients qualified for the study were divided into two groups: ‘’With ASA’’ (study group patients taking 75-mg of ASA daily for 10 years; or ‘’Without ASA’’ (control group patients who do not take ASA chronically).

All subjects in the study group were not using any other NSAID chronically. Colonoscope (CF-Q-165 L Olympus, Hamburg, Germany), biopsy forceps (FB-240 U Olympus, Hamburg, Germany), and catheter type spray (Olympus by Olympus, Hamburg, Germany) were used in the examination. All 131 subjects underwent a full colonoscopy. After the routine colonoscopy, the rectal mucosa was stained with 0.25% solution of methylene blue from the serratus line to the medial Houston’s valve. ACF was assessed using the endoscopic criteria established by Roncucci [20]. The number of ACF in the colon was divided into three groups: ACF < 5; ACF 5–10; or ACF > 10. In the statistical analysis, numerical data were presented, and real numbers, range of arrhythmic means, mean standard deviation, and results of the probability distribution. The Student’s *t*-test and the U test, chi2, were applied to determine the significance of differences in means and frequency of events in both groups. Statistica 7.1 (StatSoft, Tulusa, OK, USA) and Excel 2010 (Microsoft, Redmond, WA, USA) were used.

## 3. Results

The study group consisted of 131 patients (73 women and 58 men) with a mean age in the study group of 67 years for women and 52 years for men.

The age distribution in the groups was found to be “normal”, which enabled the evaluation of ACF incidence and characteristics according to age. The incidence of ACF in the study population correlated with the incidence of CRC in the entire population [8].

In the study group, normal ACF was found in 73 patients (*p* = 0.489), hyperplastic ACF in 49 (*p* = 0.328), dysplastic ACF in 16 patients (*p* = 0.107), mixed ACF in 11 subjects (*p* = 0.073). The study group was then divided into three subgroups according to the number of ACF observed: ACF < 5 (35 patients; 29.41%), 5–10 (70 patients; 58.82%), and >10 (14 patients; 11.76%).

## 4. Discussion

Chronic inflammation in the colon predisposes patients to colorectal carcinogenesis due to alteration in several pro- and anti-tumorigenic pathways [21]. Colon cancer is characterized by overexpression of COX-2 and elevated concentrations of Prostaglandin E2 (PGE2) and Prostaglandin F2 (PGF2). The overexpression of COX-2 is associated with inhibition of apoptosis, inhibition of terminal cell differentiation, and stimulation of angiogenesis [22]. In vitro studies show that expression of COX-2 is an early event in process of polyp formation driven by pro-inflammatory cytokines. Moreover, in the preclinical tumor model, inhibition of COX-2 decreases polyposis in mice [23]. Peiwei et al., state that the protective influence of ASA was confined to patients with positive COX-2 expression and Phosphatidylinositol 3-kinase (PI3KCA) mutation in the tumor [24]. Similarly, Chan et al., show that the protective effect of ASA was seen only in patients with COX-2 overexpression [25]. The mechanism of chemopreventive properties of ASA in ACF prevention is unknown, but it may involve the suppression of cell proliferation or the stimulation of an immune response, due to an inhibitory effect on prostaglandin synthesis. ASA inhibits the systemic Cyclooxygenase-1 (COX-1) and COX-2 activity, and therefore, colonic prostaglandin synthesis. The chronic inflammation which led to the stimulation of a local inflammatory response increased the levels of COX-2. Shiff et al., analysis of clinical trials pointed to significant influence of NSAIDs on tissue turnover processes, i.e., balance between cell proliferation and apoptosis [26]. In patients with familial adenomatous polyposis sulindac inhibits the development of adenomas, at the same time restoring normal apoptosis dynamics, which is significantly reduced in these patients [27,28].

The mechanism of anticancer action of NSAIDs is not limited to an inhibitory effect on COX activity and prostaglandin production. In recent years, several new properties in this group of drugs may be important in their suppressive effect on carcinogenesis. All tested drugs in this group inhibited the proliferation of colon cancer lineage cells, stimulating their apoptosis simultaneously. NSAIDs caused disruption of cell cycle progression and arrest in G0/G1 phase due to decreased levels of critical proteins—cyclin-dependent kinases. The other possible mechanisms include the inhibition of cyclin-dependent kinases, nuclear factor κ B signaling, activation of 5′ adenosine monophosphate-activated protein kinase (AMP-kinase) and inhibition of mammalian target of rapamycin (mTOR) signaling, inhibition of Wnt signaling and β-catenin phosphorylation, downregulation of c-Myc, cyclin A2 and cyclin-dependent kinase 2 (CDK2), induction of deoxyribonucleic acid (DNA) mismatch repair proteins, acetylation of p53, glucose-6-phosphate dehydrogenase and other proteins, and inhibition of 12-O-tetradecanoylphorbol-13-acetate induced activator protein 1 activity. In vitro studies show that NSAIDs also affect mechanisms important in cellular homeostasis regulation, such as blocking tumorigenic transformation by inhibiting Ras protein activity, and nuclear factors [29,30].

ASA has shown cardiovascular safety with long-term use compared to other NSAIDs [31]. This research aimed to examine if ASA influences rectal ACF incidences, which are the preneoplastic lesions for CRC development.

In our study, the patients taking ASA chronically (>10 years) had a lower number of all types of rectal ACFs (Table 1). The number of ACF in patients with ASA is lower in the range of 5–10, similar for ACF > 10, and higher for single ACF than in patients without ASA (Figure 1). With age, the increase in dysplastic ACF was higher in the Without ASA group (Figure 2). The With ASA group had a lower number of hyperplastic and normal ACFs than those without ASA. With age > 60, we see an increase in hyperplastic and normal ACFs in the With ASA group. In the group without ASA, there is no increase in the number of normal and hyperplastic ACF after the age of 60 (Figure 3 and Figure 4).

No mixed ACF was found in those with ASA, and after 50 years of age, the number of this type of ACF was similar in all age groups in those without ASA (Figure 5). The results of our study show results similar to other published research. Shpitz et al. [11] showed that in the ASA (100-mg) treated group there was a 64% reduction in ACF density in the proximal colon and 82% in the distal colon. In addition, there was a 48% reduction in dysplastic ACFs compared to the control group, but this difference was not statistically significant. In this study, median ASA intake was 48 months, with only 5% of subjects taking ASA longer than 10 years. Takayama et al., in a study of 189 patients after two months of treatment with sulindac (300-mg per day) and etodolac (400-mg per day) found that in polypectomy patients, the number of ACFs after two months in the sulindac group was significantly lower than in the placebo group (*p* < 0.001), while the number in the etodolac group was not significantly different from the placebo group (*p* = 0.67) [32]. Takayama et al., showed that treatment with sulindac for a year caused a decrease in rectal density of ACF [5]. Moreover, in seven subjects, the authors found atrophy of ACF, and in others, a marked reduction.

The results of our study showed that low doses of ASA can cause a decrease in the number of adenomas (Figure 6). Not taking ASA affects the occurrence of more adenomas as compared to those taking ASA. With age, the intensity of adenoma growth decreased, although there is an apparent increase in the number of adenomas in the elderly. A significant finding was that in the chronic ASA takers below 70 years of age in the study group, no adenoma was found (Table 2). Hyperplastic and serrated polyps were also found in lower numbers (Figure 7 and Figure 8). Similar data were found in the number of rectal ACFs in the study group and the control group. It is likely that dysplastic ACF, with age, transforms into macroscopically visible changes (e.g., adenomas) and their number decreases. In elderly, the rectal mucosa may atrophy, and regeneration is not as intense as in young people. ACFs, even without chemopreventive measures, can involute and disappear, but they can also transform into adenomas and CRCs [33,34,35]. When we consider the incidence of different types of polyps in patients with and without ASA in all age groups, there are similar relationships: we see an increase in the number of polyps with age, with a decrease in the rate of polyp growth (Figure 6, Figure 7 and Figure 8).

Baron et al., show that the reduction in the risk of advanced adenomas (compared with placebo) was significant: more than 40% in the group received 81-mg of ASA daily [36]. The researchers suggest that ASA may have a stronger effect in the later stages of conversion of nonadvanced adenomas to advanced adenomas. A study by Sandler et al., indicated that several years of ASA use can reduce adenoma recurrence in patients with a history of adenoma and CRC [37]. Similar findings were reported by Chan et al. [38] where among more than 27,000 women studied, 1368 were diagnosed with colorectal adenoma. The authors found the greatest protective effect at substantially higher doses (>14 tablets/week). In contrast to dose escalation, increasing the duration of ASA use did not reduce the risk of adenoma. Other research also confirms this data [39,40].

A marked reduction in adenomas after six months of treatment with sulindac was shown by Steinbach et al. [41]. Their study included patients with familial adenomatous polyposis (FAP). Treatment with twice daily 400-mg celecoxib resulted in significant reductions in colorectal polyps after a six-month study.

A study by Sinicrope et al., found that the combination of ASA with the polyamine synthesis inhibitor, difluoromethylornithine, did not significantly reduce adenoma recurrence at the one-year follow-up (28.8% adenoma in the DFMO group and 44.8% adenoma in the placebo group) [10].

Many chemoprevention studies use different doses of ASA (from 81-mg to 325-mg). Currently, there are no recommendations for ASA dose multiplicity in such studies. Low doses of ASA, as determined by Ruffin et al., to be 81-mg, significantly reduce PGE2 and PGF2 levels in the colonic mucosa [42]. Other authors used higher doses, but the differences in the chemopreventive effect were not statistically significant [43]. Additionally, Patrono et al., suggested that higher dose ASAs are needed to inhibit the COX-2 isoenzyme [44].

Other experimental studies suggest that ASA preparations may also act through mechanisms other than COX-2 and may require higher doses [45,46]. It is not currently, nor fully, explained why lower doses of ASA (81-mg) have similar effects on ACF and colorectal adenoma incidence as higher doses (325-mg). It is important to remember when using ASA doses, higher doses may inhibit the production of PGE2 and other prostaglandins with protective effects on the colonic lining membrane. The damage of the mucosa and loss of the mucosal barrier exposes the deeper layers of the colon wall, thereby exposing them to bile acids, fatty acids, and toxic bacterial products that further damage the intestinal barrier, which potentiates the toxic and carcinogenic effects of various chemicals present in the feces [47]. Grau et al., in a study of 850 patients demonstrated a protective effect of low-dose ASA (81-mg) similar to Ruffin et al., on adenomas at the four-year follow-up [42,48]. The risk of adenoma after four years was lower in the ASA group (26.8%) compared to the placebo group (39.9%). The results for subjects taking 325-mg ASA were similar, although not statistically significant. Two clinical studies have also been published that do not support a significant effect of ASA preparations on CRC incidences [33,49]. Perhaps more important than the dose of ASA itself is the duration of treatment [39]. This may warrant further investigation. Dulai et al., in meta-analysis, show that high doses of ASA have the same efficacy in CRC prevention as low doses, but a greater risk of side effects. Several studies were conducted to establish the clinical value of ASA; however, it differs in terms of dose (81, 100, 160, 325-mg) and primary-end point measure, which may be one of the reasons for inconclusive results [50]. There are many studies which show a chemopreventive effect of low doses of ASA on the incidence of ACF, colorectal adenomas, and CRC [32,33,36,38,40]. There are also reports, such as the work of Maxon et al. and Rudolph et al., which do not confirm the reduction in the number of ACF after treatment with low-dose, non-steroidal, anti-inflammatory drugs [51,52]. The limitations of those studies were the small study group and the low frequency of use of the drug (at least twice a week). The other reason for differences between the studies may be the ethnicity of the study population. In the Maxon et al., study, 73.5% were African American, 18% were Caucasian/White, and 8.5% were Latin American. In our study, all the subjects were Caucasian/White. There are other factors which potentially influence ACF formation including smoking, age, and diet [51,52,53,54,55]. In one of our previous studies, we show that the age when patients start chemoprevention is one of the most important factors [53]. Therefore, the limitation of the study group includes the low number of patients, especially in the younger patients group.

The new hypothesis regarding the ACF chemoprevention is to use NSAIDs in combination with other drugs. Sinicrope et al., showed that difluoromethylornithine (DFMO), a polyamine synthesis inhibitor, works synergistically with NSAIDs in the chemoprevention of colorectal cancer [10]. In a study of 104 patients, the researchers showed a statistically significant reduction in rectal ACFs after one year of DFMO (500-mg once daily) plus ASA (325-mg once daily). The combination of low doses of DFMO and ASA administered continuously for one year did not result in a statistically significant reduction in colorectal adenoma recurrence in this group. The study by Sinicrope et al., showed regression of ACFs after DFMO and ASA. This research was further confirmed in a study by Li et al., which was conducted on an azoxymethane model of colon cancer in rodents [56]. The chemopreventive effect of ASA was also studied in combination with metformin in patients with rectal ACFs [57]. Honoso et al., concluded that low-dose metformin (250-mg/day) was safe and inhibited the formation of colorectal ACFs [58]. In a randomized clinical trial, Higurashi et al., showed that low-dose metformin was safe and reduced the incidence of new polyps in patients after colorectal polypectomy [59].

## 5. Conclusions

Patients who take chronic (>10 years) low doses of ASA have a lower total number of all types of rectal ACFs, and adenomas compared to patients who do not take ASA.

## Figures and Tables

**Figure 1 medicina-58-01767-f001:**
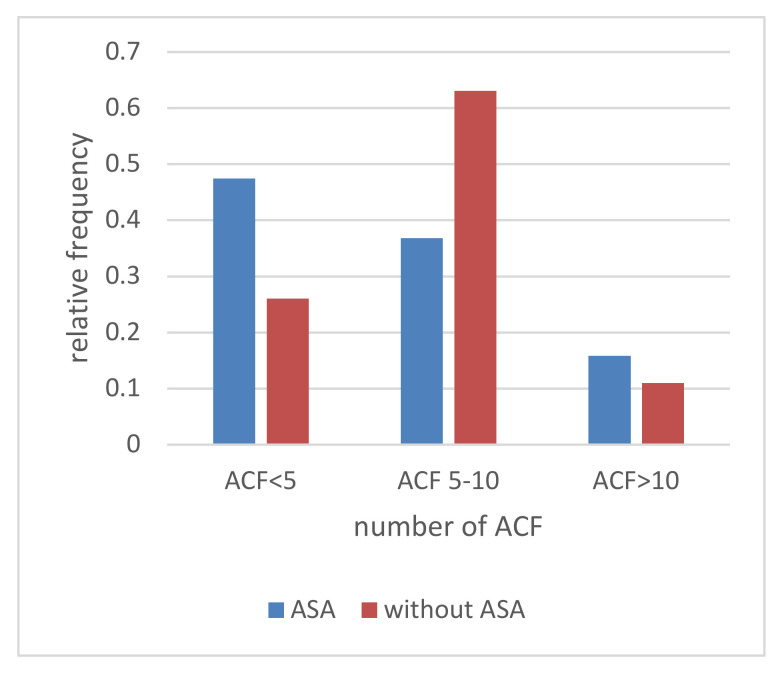
Relationship of the incidence of rectal aberrant crypt foci (ACF) counts to with ASA or without ASA. The rate of ACF formation in patients without ASA is higher than in patients with ASA. There is a difference in the probability distributions of the occurrence of ACF count in patients with and without ASA and it is statistically significant according to the X2 criterion with a significance level of *p* < 0.05.

**Figure 2 medicina-58-01767-f002:**
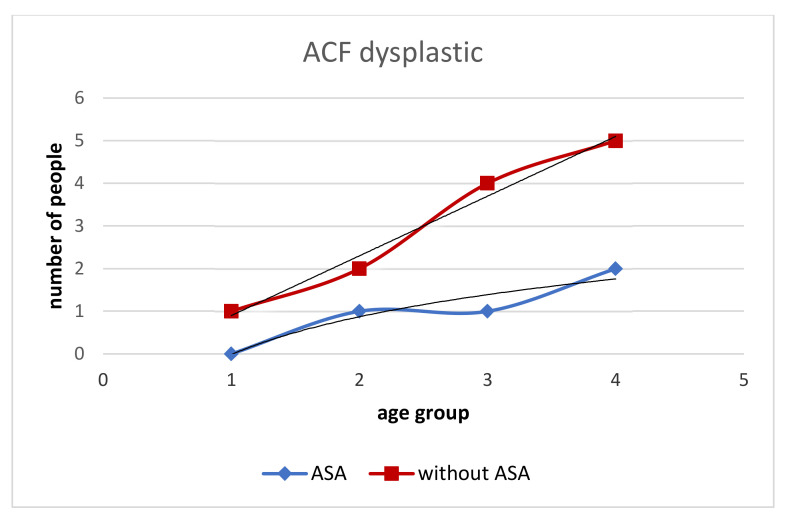
Incidence of dysplastic ACFs in the group with ASA and without ASA. The total number of dysplastic ACFs is higher in those without ASA. The increase in the number of dysplastic ACFs decreases with age in both groups (with ASA and without ASA), with the increase being greater in those without ASA.

**Figure 3 medicina-58-01767-f003:**
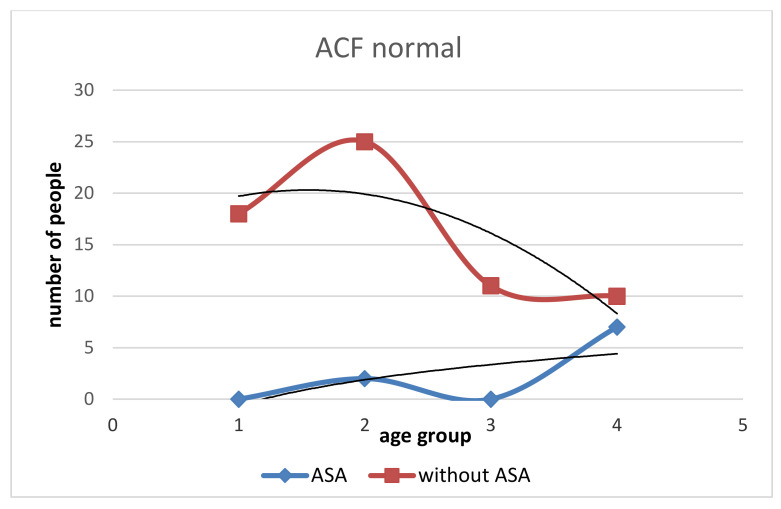
Incidence of ACF normal in the group with ASA and without ASA. In those without ASA, the number of ACF normally increases rapidly in the age group 50–60 years, then decreases from age 60 years. In the group with ASA, there is a lower number of ACF normal.

**Figure 4 medicina-58-01767-f004:**
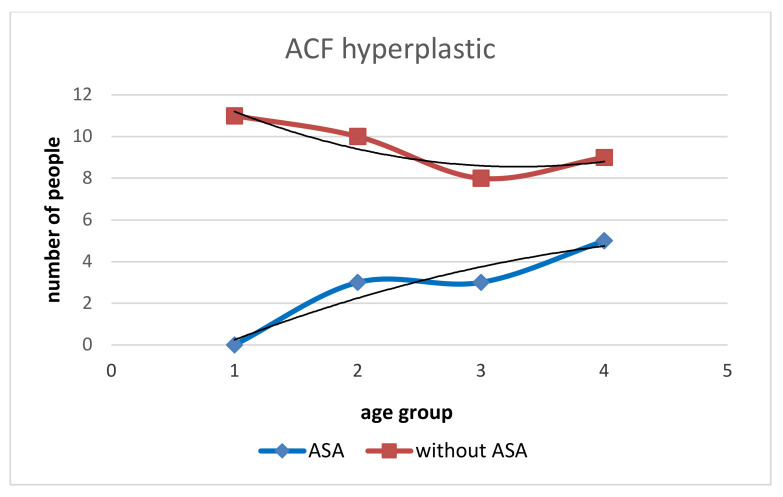
Incidence of hyperplastic ACF in the group with ASA and without ASA. The number of hyperplastic ACFs in subjects without ASA is greater than those with ASA. In subjects without ASA, the number and increment of hyperplastic ACFs gradually decrease with age. In subjects with ASA, the number and increment of hyperplastic ACFs increases with age. The difference in the number of hyperplastic ACFs between with ASA and without ASA decreases with age, but even after age 70 there are more in those without ASA.

**Figure 5 medicina-58-01767-f005:**
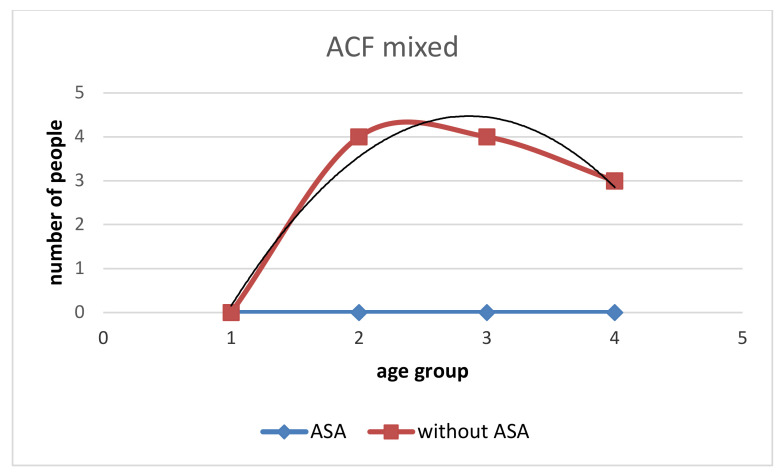
Incidence of ACF mixed in the group with ASA and without ASA. No ACF mixed was observed in subjects with ASA. In those without ASA, the increase in the number of ACF mixed decreases with age. After age 50, the number of this type of ACF is similar in all age groups in those without ASA.

**Figure 6 medicina-58-01767-f006:**
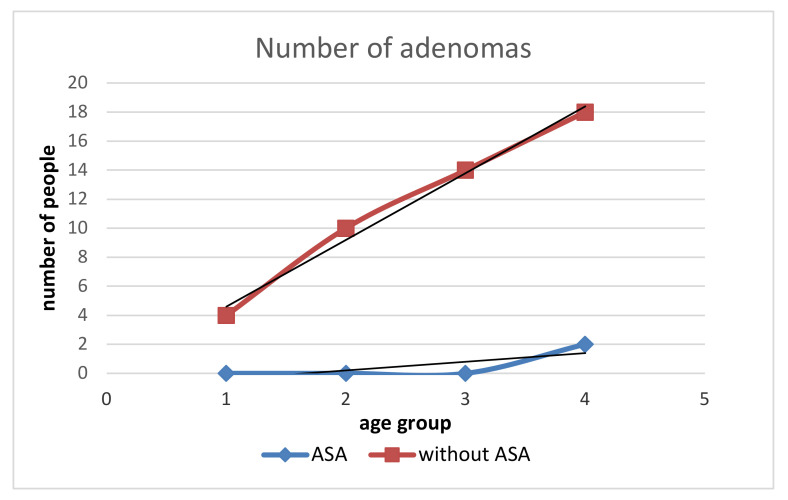
Relationship of adenoma incidence in subjects with ASA and without ASA across age groups. With age, the intensity of adenoma growth decreases, although there is an apparent increase in the number of adenomas in the elderly. Without ASA affects the incidence of more adenomas compared to those with ASA. The curve of adenoma incidence in subjects without ASA can be described by the equation y = 4.6x with a coefficient of determination of 0.98.

**Figure 7 medicina-58-01767-f007:**
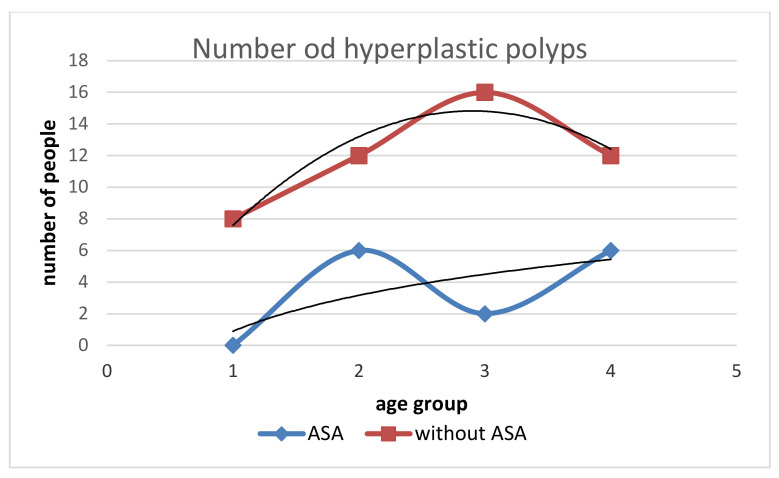
Relationship of hyperplastic polyp incidence in people with ASA and without ASA by age groups. The number of hyperplastic polyps in people without ASA increases between the ages of 50–70 years and then decreases. The rate of growth of hyperplastic polyps in people without ASA decreases with age. In people taking ASA, the number of hyperplastic polyps increases in the age group of 50–60 years and then decreases to take an increasing trend again from the age of 70 years. The trend of decreasing rate of growth of these polyps is similar to that in people without ASA.

**Figure 8 medicina-58-01767-f008:**
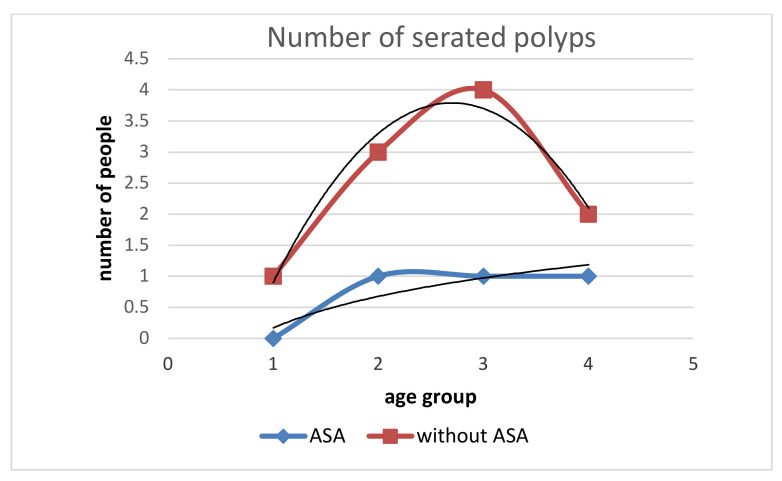
Relationship of serrated polyp incidence in subjects with ASA and without ASA by age groups. In subjects without ASA, the growth rate and total number of serrated polyps decreases with age. There is no increase in the number of serrated polyps in people with ASA between the ages of 50–70 years.

**Table 1 medicina-58-01767-t001:** Occurrence of ACF types in age groups in the with ASA and without ASA groups.

Age Group	Type of Aberrant Crypt Foci							
	Normal		Hyperplastic		Dysplastic		Mixed	
	With ASA	Without ASA	With ASA	Without ASA	With ASA	Without ASA	With ASA	Without ASA
1. <50	0	18	0	11	0	1	0	0
2. 50–60	2	25	3	10	1	2	0	4
3. 60–70	0	11	3	8	1	4	0	4
4. >70	7	10	5	9	2	5	0	3
Total	9	64	11	38	4	12	0	11

**Table 2 medicina-58-01767-t002:** Occurrence of polyps in age groups for the with ASA and without ASA groups.

Age Group	Adenoma		Hyperplastic Polyp		Serated Polyp	
	With ASA	Without ASA	With ASA	Without ASA	With ASA	Without ASA
1. <50	0	4	0	8	0	1
2. 50–60	0	10	6	12	1	3
3. 60–70	0	14	2	16	1	4
4. >70	2	18	6	12	1	2
Total	2	46	14	48	3	10

## Data Availability

Data availability for request from the authors.

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
