# Peer review of "The Effect of Low Doses of Acetylsalicylic Acid on the Occurrence of Rectal Aberrant Crypt Foci"

_medicina, 2022, doi:10.3390/medicina58121767_

Round 1
Reviewer 1 Report
Although this work is of interest, however, I have some concerns about it and my observations are reported below:
In the Results section, the presentation of data needs improvement. Only Figures and Tables are provided, often unclear (see lines 150-151 in discussion section) and sometimes incorrect in the text.
Authors say: "The rate of ACF formation in patients in the"Without ASA" group was higher than that in patients taking ASA (Fig.1). With age, the increase in dysplastic ACF was higher in the "Without ASA" group (Fig.2), while for normal and hyperplastic ACF was similar in both groups (Fig.3,4)".
Checking Fig. 1, their affermation is true only for ACF 5-10. In addition, comparing Fig. 3 and 4, they don’t look similar.
In the Discussion section, results should be described extensively.
Minor remarks:
Each abbreviation should be explained when used for the first time (see PGE2 and PGF2, line 134).
Standardize the name used for ASA; Aspirin or acetyl salicylic acid.
I recommend the Authors to continue their interesting work also improving the English language.
Author Response
Although this work is of interest, however, I have some concerns about it and my observations are reported below:
In the Results section, the presentation of data needs improvement. Only Figures and Tables are provided, often unclear (see lines 150-151 in discussion section) and sometimes incorrect in the text.
Authors say: "The rate of ACF formation in patients in the without ASA group was higher than in patients with ASA (Fig.1). With age, the increase in dysplastic ACF was higher in the without ASA group (Fig.2). However, normal and hyperplastic ACF was similar in both groups (Fig.3,4)".
Checking Fig. 1, their affirmation is true only for ACF 5-10. In addition, comparing Fig. 3 and 4, they don’t look similar.
In the Discussion section, results should be described extensively.
Minor remarks:
Each abbreviation should be explained when used for the first time (see PGE2 and PGF2, line 156).
Standardize the name used for ASA; Aspirin or acetyl salicylic acid.
Authors say: Statements have been made in both the abstract and the body of the text to clarify the use of ASA. As well as the subsequent use of aspirin in the literature review and discussion.
I recommend the Authors to continue their interesting work also improving the English language.
Authors say: We thank the reviewer for constructive comments and suggestions. Below please find our response to the concerns raised:
- We improved the presentation data of figures and tables according to your suggestion.
- We changed the Fig 2. description.
- Ane abbreviation section was added to the text and the name of the drug was standardized to use the acetylsalicylic acid. Additionally, a statement was added to both the abstract and the text explaining the abbreviation, as mentioned above. All the abbreviations were explained in the text.
- The discussion was altered to read:
In our study, the patients taking ASA chronically (>10 years) had a lower number of all types of rectal ACFs (Table 1). The number of ACF in patients with ASA is lower in the range of 5-10, similar for ACF >10 and higher for single ACF than in patients without ASA.(Fig.1). With age, the increase in dysplastic ACF was higher in the Without ASA group (Fig.2). The with ASA group had a lower number of hyperplastic and normal ACFs than those without ASA. With age> 60, we see an increase in hyperplastic and normal ACFs in the with ASA group. In group without ASA, there is no increase the number of normal and hyperplastic ACF after the age of 60 (Fig. 3,4).
W grupie przyjmującej ASA występuje mniejsza liczba ACF hiperplastycznych i normalny w stosunku do nie przyjmujących ASA. Wraz z wiekiem >60 rok życia widzimy wzrost liczby ACF hiperplastycznych i normalnych w grupie przyjmujących ASA. U osób bez ASA nie stwierdzamy przyrostu liczby ACF normalnych i hiperplastyczny po 60 roku życia(Fig.3,4)
Reviewer 2 Report
I appreciate the opportunity to review this interesting research manuscript Title: The effect of low doses of Aspirin on the occurrence of rectal 2 aberrant crypt foci (ACF).
In this study, authors demonstrated the effect of low-dose aspirin and interactions with ACF in colorectal cancer patients. The experiment design and the presentation of data are good. The manuscript was well-analyzed, and I am satisfied with the figures. There is a clear effort in the work. However, several points should be addressed by the authors. This research might have its clinical significance but still, but improvements are required. The recommendations for the manuscript are explained below.
Comment 1: The authors should define the abbreviations for the first time in the abstract, text of the manuscript, and figure legends, and then follow with the abbreviations in the whole of the manuscript.
Comments 2: In my opinion, the introduction section could not show the gap in science well in this research. The authors need to improve it.
Comments 3; Please go over your manuscript text and ensure it is written in an acceptable English language.
Comments 4:Authors must clear the problem statement and objectives in the abstract
Comments 5: the last paragraph of the Introduction needs clear and well-defined objectives for the study
Comment 6: Results from this large-scale, long-term trial suggest that alternate-day use of low-dose aspirin (100 mg) for an average of 10 years of treatment does not lower the risk of total, breast, colorectal, or other site-specific cancers, how could authors justify this statement in your manuscript?
Comments 7: The doses of aspirin that are associated with lower colorectal cancer risk have been as low as 81 mg of aspirin a day
Comments 8: Do the authors have a special reason for choosing the Aberrant crypt foci (ACF)?
Comments 9: do the effect of lower dose aspirin show the same level of effect in both males and females and younger and elderly peoples
Comments 10:Authors must explain the mechanistic differences in the effect of aspirin in preclinical models,
Comments 11:Authors must include a brief explanation about the association of ACF with ASA concerning function and effect
Comment 12:In the result section, the authors should define the abbreviations of each term.
Comment 13: why does the effect of dysplastic ACFs decrease with age in both groups
Comment 14: In conclusion, the Authors must mention the limitation of this study and the future perspectives of the studies
Comment 15 The quality of some pictures is poor. Some word is hard to recognize. The authors need to revise that.
Comments 16 It is suggested to check the manuscript for English grammar once more. Also, the authors have written very short sentences or non-academic sentences in the manuscript; therefore, they need to correct and edit with professional English editing.
Author Response
Dear Reviewer,
kindly thank you for your accurate comments and the time spent on improvements to improve our manuscript.
We applied all your advice.
I appreciate the opportunity to review this interesting research manuscript Title: The effect of low doses of Aspirin on the occurrence of rectal 2 aberrant crypt foci (ACF). In this study, authors demonstrated the effect of low-dose aspirin and interactions with ACF in colorectal cancer patients. The experiment design and the presentation of data are good. The manuscript was well-analyzed, and I am satisfied with the figures. There is a clear effort in the work. However, several points should be addressed by the authors. This research might have its clinical significance but still, but improvements are required. The recommendations for the manuscript are explained below.Authors say: Thank you for the being interested in our study and constructive comments and suggestions. Below please find our response to the concerns raised:
Comment 1: The authors should define the abbreviations for the first time in the abstract, text of the manuscript, and figure legends, and then follow with the abbreviations in the whole of the manuscript.
The abbreviations section was added to the text. Additionally, the use ASA and aspirin in both the abstract and the text has been addressed in both locations.
Comments 2: In my opinion, the introduction section could not show the gap in science well in this research. The authors need to improve it.
As recommended by the reviewer, the introduction from lines 54 to 66 was modified and corrected.
Comments 3; Please go over your manuscript text and ensure it is written in an acceptable English language.
The use of the English language was checked by native speaker.
Comments 4: Authors must clear the problem statement and objectives in the abstract
As recommended by the reviewer, the description of the problem and objectives were modified and changed in the abstract to clear the problem statement and objectives of the study, lines 17-21
Comments 5: the last paragraph of the Introduction needs clear and well-defined objectives for the study
An objective of the study to the manuscript.
Comment 6: Results from this large-scale, long-term trial suggest that alternate-day use of low-dose aspirin (100 mg) for an average of 10 years of treatment does not lower the risk of total, breast, colorectal, or other site-specific cancers, how could authors justify this statement in your manuscript?
We add the following new information to the article (line 271-287)
Perhaps more important than the dose of ASA itself is the duration of treatment [39]. This may warrant further investigation. Dulai et al., in meta-analysis, show that high doses of aspirin have the same efficacy in CRC prevention as low doses, but greater risk of side effects. Several studies were conducted to establish the clinical value of aspirin, however it differs in terms of dose (81-mg, 100-mg, 160-mg, 325-mg) and primary-end pint measure, which may be one of the reason of inconclusive results [50]. There are many studies which show a chemopreventive effect of low doses of ASA on the incidence of ACF, colorectal adenomas, and CRC [32,33,36,38,40]. There are also reports, such as the work of Maxon et al. and Rudolph et al., which do not confirm the reduction in the number of ACF after treatment with low-dose, non-steroidal, anti-inflammatory drugs [51,52]. The limitations of those studies were the small study group and the low frequency of use of the drug (at least twice a week). The other reason for differences between the studies may be the ethnicity of the study population. In Maxon et al. study 73.5% were African American, 18% were Caucasian/White, and 8.5% were Latin American. In our study all the subjects were Caucasian/White. There are other factors which potentially influence ACF formation including smoking, age, and diet [51-55]. In one of our previous studies, we show that the age when patients starts chemoprevention is one of the most important factor [53]
Comments 7: The doses of aspirin that are associated with lower colorectal cancer risk have been as low as 81 mg of aspirin a day
Currently, there are no established minimum doses of acetylsalicylic acid that show a chemopreventive effect in the formation of ACF and adenoma. This information was added in lines 254-269.
Comments 8: Do the authors have a special reason for choosing the Aberrant crypt foci (ACF)?
With , colorectal cancer (CRC) remaining as one of the most common causes of cancer deaths worldwide, the first author of this study (Kowalczyk) has been dealing with gastrointestinal endoscopy for 20 years and the problem of ACF, which has fascinated him for years. As such, the authors are planning a new study focusing on ACF soon.
Comments 9: do the effect of lower dose aspirin show the same level of effect in both males and females and younger and elderly peoples
In a recent clinical study, gender differences in the pharmacokinetics of salicylates (DOI: 10.1007 / BF00542358) may be clinically significant because men achieved lower plasma aspirin levels than women when following the same weight-adjusted dose of ASA. However, we did not address this in the current study. Regarding the age of the patients, the study by Chuan-Guo et al. DOI: 10.1001 / jamaoncol.2020.7338) showed that initiation of aspirin at an older age (≥70 years) was not associated with lower risk of CRC. In contrast, those who initiated aspirin at a younger age and continued use as they aged appeared to derive continued benefit of CRC risk reduction. Taken together with the results of the ASPREE trial, these findings suggest that initiation of aspirin use at an older age for the sole purpose of primary prevention of CRC should be discouraged. However, our findings appear to support recommendations to continue aspirin use if initiated at a younger age. Further studies to elucidate biologic mechanisms of aspirin according to age are warranted.
Comments 10: Authors must explain the mechanistic differences in the effect of aspirin in preclinical models,
We add the suggested information to the manuscript. The chemopreventive effect is presented in lines 181 to 190
Comments 11:Authors must include a brief explanation about the association of ACF with ASA concerning function and effect
The mechanism of action of ASA is explained in lines 181-190 The possible effect of ASA, in relation to the formation of ACF, is described per the reviewer's recommendation in lines 54-62.
Comment 12:In the result section, the authors should define the abbreviations of each term.
Each term and abbreviation has been defined in the manuscript. Additionally, we have added an abbreviation section for reference.
Comment 13: why does the effect of dysplastic ACFs decrease with age in both groups
A comment has been added to our manuscript:
It is likely that dysplastic ACF with age transforms into macroscopically visible changes (e.g., adenomas) and their number decreases. In elderly the rectal mucosa may atrophy, and regeneration is not as intense as in young people.
Comment 14: In conclusion, the Authors must mention the limitation of this study and the future perspectives of the studies
We added the limitations of the study to the discussion section.
Comment 15 The quality of some pictures is poor. Some word is hard to recognize. The authors need to revise that.
The quality of figures has been improved.
Comments 16 It is suggested to check the manuscript for English grammar once more. Also, the authors have written very short sentences or non-academic sentences in the manuscript; therefore, they need to correct and edit with professional English editing.
We believe we have improved the quality of the English grammar and the quality of work. The manuscript was proofread and edited y native English speaker. The non-academic sentences were replaced.
Reviewer 3 Report
This work compared the incidence of rectal ACFs and colorectal adenomas in a group of patients taking acetylsalicylic acid(ASA) and revealed chronic 28 (>10 years) low doses of ASA have a lower total number of all types of rectal ACFs and adenomas. The comments are as below:
1、 The horizontal lines in the Figures 1-4 should be avoided.
2、 The text of vertical coordinate was overplayed with the numbers.
3、 Tables 1 and 2 should be modified to three-line table.
4、 Line 125: Please add and before pointed out.
5、 Line 141: the sentence should be rephrased.
Author Response
Dear Reviewer,
kindly thank you for your accurate comments and the time spent on improvements to improve our manuscript.
We applied all your advice.
This work compared the incidence of rectal ACFs and colorectal adenomas in a group of patients taking acetylsalicylic acid(ASA) and revealed chronic 28 (>10 years) low doses of ASA have a lower total number of all types of rectal ACFs and adenomas. The comments are as below:
1、 The horizontal lines in the Figures 1-4 should be avoided.
Authors say: Thank you for your thoughtful review and comments. Our comments regarding the edits are discussed below.
We corrected the horizontal lines, in accordance to reviewer recommendation.
2、 The text of vertical coordinate was overplayed with the numbers.
We corrected the vertical coordinates and the numbering, as requested.
3、 Tables 1 and 2 should be modified to three-line table.
We corrected Tables 1 and 2, in accordance to reviewer recommendation.
4、 Line 125: Please add and before pointed out.
We corrected Line 125.
5、 Line 141: the sentence should be rephrased.
As per the reviewer’s request and suggestion, the sentence at Line 141 has been rephrased.
Round 2
Reviewer 1 Report
The manuscript has been sufficiently improved and can be considered for the publication.